# The Association of Meteorological Factors with Cognitive Function in Older Adults

**DOI:** 10.3390/ijerph18115981

**Published:** 2021-06-02

**Authors:** Yuehong Qiu, Kaigong Wei, Lijun Zhu, Dan Wu, Can Jiao

**Affiliations:** 1School of Psychology, Shenzhen University, Shenzhen 518052, China; 1950482005@email.szu.edu.cn (Y.Q.); 2070481006@email.szu.edu.cn (K.W.); zhulijunjxnu@163.com (L.Z.); wudan.tracy@szu.edu.cn (D.W.); 2Institute of Mental Health, Shenzhen University, Shenzhen 518052, China

**Keywords:** older age, cognitive function, meteorological variables, mixed effects model

## Abstract

Individual and meteorological factors are associated with cognitive function in older adults. However, how these two factors interact with each other to affect cognitive function in older adults is still unclear. We used mixed effects models to assess the association of individual and meteorological factors with cognitive function among older adults. Individual data in this study were from the database of China Family Panel Studies. A total of 3448 older adults from 25 provinces were included in our analysis. Cognitive functions were measured using a memory test and a logical sequence test. We used the meteorological data in the daily climate dataset of China’s surface international exchange stations, and two meteorological factors (i.e., average temperature and relative humidity) were assessed. The empty model showed significant differences in the cognitive scores of the older adults across different provinces. The results showed a main impact of residence (i.e., urban or rural) and a significant humidity–residence interaction on memory performance in older adults. Specifically, the negative association between humidity and memory performance was more pronounced in urban areas. This study suggested that meteorological factors may, in concert with individual factors, be associated with differences in memory function in older adults.

## 1. Introduction

The aging population has rapidly increased worldwide. By 2050, the global population aged 60 years and older is expected to total 2 billion, up from 900 million in 2015 [1]. Normal or pathological (e.g., mild cognitive impairment and Alzheimer’s disease) cognitive decline associated with aging is often accompanied by reduced quality of life [2,3,4]. Moreover, cognitive impairment in older age has caused substantial financial and societal burdens. For example, direct and indirect costs for individuals with cognitive impairment have reached nearly $244 billion annually worldwide [5]. Thus, a better understanding of the factors that influence the speed of the cognitive decline over aging is warranted.

Multiple factors have been associated with cognitive function, such as age, gender, educational and nutritional level, socio-economic status, physical activity/sedentary behaviors, and chronic disease conditions. Some researchers have investigated the preventable effects of lifestyle behaviors (e.g., physical activity and prolonged sitting) on cognitive function in older age, showing that adopting healthy behaviors reduced the speed of the cognitive decline across aging [6,7,8]. In addition to these individual and lifestyle risk factors, meteorological and environmental factors have also attracted scientific attention.

Previous studies have shown that environmental factors such as urbanization and traffic (e.g., highway construction) have been associated with cognitive function in older age [9,10,11,12]. For example, living in cities appears to reduce the risk of cognitive impairment, although the exact reasons are unclear [12,13]. This is not only because of the individual factors (e.g., higher occupational complexity) of urban elderly, notably, it has been suggested that the effect of this environmental factor on cognitive function can be explained by its influence on microclimate and ecology. Indeed, studies have shown that meteorological factors such as precipitation, sunshine, and other natural phenomena related to atmospheric movement may influence cognitive function [14,15,16]. In particular, among these multiple meteorological indicators, temperature (i.e., air temperature) and humidity have been highlighted because they are thought to represent the most stable, common, and sensitive meteorological dimensions [17,18]. Other meteorological indicators such as ground temperature and atmospheric pressure are essentially following the variation of temperature and humidity, so they have serious multicollinearity problems. Therefore, it is meaningful to explore the influence of temperature and humidity on cognitive function in older age.

However, evidence of the associations of temperature and humidity with age-related cognitive decline is mixed. Some studies suggest that extreme (either cold or heat) temperature influences the rate of cognitive decline across aging (e.g., alertness, decision-making ability, and memory) [19,20]. However, other studies find no evidence of an association between temperature and cognitive function [21,22]. Moreover, some studies point out that humidity is a risk factor for the physiological and cognitive function in old age [23,24], but related research is still lacking. This lack of consistency may result from various factors. First, most studies have only investigated the influence of a single meteorological factor on cognitive function, which may have biased the associations observed. Second, study often relied on small sample sizes (n < 30) [21,22]. Third, the statistical approach used in previous studies (i.e., traditional regression model, which ignores the nesting of data, greatly reducing the practical significance of the research results) was not particularly suited to accurately estimate the associations between meteorological factors (i.e., a macro level factor) and cognitive functions (a micro level factor), while accounting for individual factors. It has been pointed out in the literature that meteorological factors, such as temperature and humidity, have different effects on demographic factors (e.g., gender and age) [25,26,27]. Therefore, it is necessary to explore the interaction between meteorological factors and individual factors.

To fill this knowledge gap, the objective of the current study was to investigate, using an analytical strategy suited to examine the multiple levels of influence (i.e., micro and macro), the associations of temperature and relative humidity with the level of and change in cognitive function across aging. We used the monthly average temperature and relative humidity of each province in China to measure the meteorological factors. Cognitive functions were measured using cognitive tests. Age, gender, residence (urban or rural), and chronic disease (e.g., diabetes and hypertension) in older age were used as individual variables. In addition, as mentioned above, the traffic level influences people’s cognitive function through its impact on microclimate and ecology [9,10], and therefore, we use highway construction as a covariate.

We used a mixed model, as shown in Figure 1, to test the hypotheses that the variables at the cross-level interactions had main and moderating effects on cognitive function in older age. We hypothesized that individual factors would be associated with cognitive function level and change across aging (H1a and H1b). Moreover, we hypothesized that temperature and humidity would be associated with cognitive function level and change across aging (H2a and H2b). Finally, we hypothesized that the association between individual factors and cognitive functions’ level of and change across aging would be moderated by temperature and humidity (H3a and H3b).

## 2. Methods

### 2.1. Data Sources and Participants

In this study, data were extracted from the China Family Panel Studies (CFPS), a national large-scale screening dataset with provinces (or autonomous regions/municipalities) as the unit. Cognitive function was assessed in 2016 using a battery of tests, including a memory test and a logical sequence test. Geographic location and interview date were used to accurately match cognitive scores with the local meteorological data during the month cognitive tests were administered. Specifically, the meteorological data in this study were from the daily climate dataset of China’s surface international exchange stations (Surface_Climate Data_China_ Multiple Elements_Daily Value Data_Climate Exchange Station, SCCMDC, V3.0). The dataset contained daily records of meteorological conditions from 166 monitoring stations in China. As the covariate, highway data in traffic came from the *China Statistical Yearbook* of 2017, the data of which are compiled by the National Bureau of statistics of China (http://www.stats.gov.cn, 22 February 2021).

We included data for participants aged 60 years or above, who provide demographic information including gender, age, residence, years of education (years of education ≤6 means having received basic education, excluding illiteracy, and >6 means reaching the level of junior high school or above) and chronic diseases (excluding mild cognitive impairment and different forms of dementia), and who completed the cognitive tests. We excluded participants who did not have information on the interview date and on their provinces of living. A total of 3448 older adults from 25 provinces (or autonomous regions/municipalities) participated in the study. The demographic information of all the participants is shown Table 1.

### 2.2. Variable Measurement

#### 2.2.1. Individual Level Variables

The individual level variables were age, gender, residence (i.e., urban or rural), and chronic disease. The information was extracted from the demographic information recorded in the CFPS. In this study, gender, residence, and chronic diseases were used as categorical variables, and age was used as a continuous variable.

#### 2.2.2. Provincial Level Variables

At the provincial level, we mainly focused on two meteorological variables, monthly average temperature and relative humidity in 2016. The statistical regulations were as follows: (1) the daily average temperature and relative humidity were the average values of four time observations (2:00 a.m., 8:00 a.m., 2:00 p.m., 8:00 p.m.) per day for a month; (2) when a certain fixed time value was not measured, the daily average value of corresponding elements was not measured.

The hourly observation data from the ground automatic station uploaded in real time had passed the station quality control. The National Meteorological Information Center of China inputs the observation data and station quality control codes uploaded in real time into the real-time database for data service (http://data.cma.cn, 9 September 2020).

We matched provincial level meteorological data with CFPS samples in the following way. First, we extracted the month and the location (on the provincial level) the participants were interviewed from the CFPS. Second, the monthly average values of temperature and humidity of each province in the corresponding month were extracted from SCCMDC. Finally, meteorological data were matched with the cognitive scores of subjects in CFPS according to dates and provinces.

#### 2.2.3. Outcome Variables

The memory test and the logical sequence test in CFPS were used to measure cognitive function. The prototypes of the two tests are from the health and retirement study (HRS). In the memory test, the interviewer read 10 common words (such as mountain, rice, river, etc.) to the participant. After hearing all 10 words, the participants recalled the words read by the interviewer. The score of this recall was called the immediate recall score. After a few minutes, the interviewer would ask the participant to recall the 10 words again. The score of this recall was called the delayed recall score. The score of the memory test was the total number of words correctly answered by the older adult, and the order was not required.

The logical sequence test used two stages of an adaptability test. This design was based on the item response theory (IRT) [28]. In the first stage, the interviewees answered three series of questions, and the number of correct answers (0 to 3) in the first stage was obtained. In the second stage, the system selected the corresponding group of questions from four groups according to the number and distributed them to the interviewees. The difficulty of four groups of questions was graded, in the first stage, individuals with more correct answers would receive more difficult tests in the second stage. The score calculated by the Rasch model was directly provided in CFPS (http://www.isss.pku.edu.cn/cfps, 16 August 2020).

#### 2.2.4. Covariate

Highway construction as a provincial level was used as a covariate. It refers to the total length of public transit (such as bus lanes, etc.) and rail transit in a province. We used the *China Statistical Yearbook* of 2017, the data of which were compiled by the National Bureau of Statistics of China, to calculate the average highway mileage of 25 provinces, and matched traffic data according to the matching principles of the meteorological data and CFPS individual data.

### 2.3. Data Analysis

In this study, a mixed effect model (MEM) was used to evaluate the effects of different level variables on cognitive performance of older adults. Statistical assumptions associated with MEM (i.e., normality of the residuals, linearity, multicollinearity, and undue influence) were checked and were met for all models (Appendix A Appendix A, Appendix A).

The cognitive score (the memory test and logical sequence test were analyzed separately) of CFPS was used as the outcome variable ($Y_{ij}$). In accordance with previous studies [29,30], the model was established by the following steps.

Firstly, an empty model was established to test whether there was significant difference in cognitive scores of the older adults in different provinces. The total variance of the outcome variable was divided into two parts, the between-province variance ($\upsilon_{0j}$) and within-province variance ($\varepsilon_{ij}$). Equation (1) indicated that the cognitive scores of each older adult individual ($Y_{ij}$) could be estimated by the average cognitive scores of the older adults in all provinces ($\gamma_{00}$), between-province variance ($\upsilon_{0j}$), and within-province variance ($\varepsilon_{ij}$).
(1)$$\;\;\;\;\;\;\;\;\;\;\;\;\;\;\;\;\;\;\;\;Y_{ij}=\gamma_{00}+\upsilon_{0j}+\varepsilon_{ij}$$

Secondly, we established an individual level model to test the influence of individual level variables on cognitive scores. As shown in Equation (2), we added four variables at the individual level. The cognitive scores of each older adult individual ($\gamma_{ij}$) could be estimated by the average cognitive scores of the older adults in all provinces ($\gamma_{00}$), age ($\gamma_{10}$), gender ($\gamma_{20}$), residence ($\gamma_{30}$), and chronic disease ($\gamma_{40}$), and between-province variance ($\upsilon_{0j}$) and within-province variance ($\varepsilon_{ij}$).
(2)$$Y_{ij}=\gamma_{00}+\gamma_{10}age_{j}+\gamma_{20}gender_{j}+\gamma_{30}residence_{j}+\gamma_{40}chronic\;disease_{j}+\upsilon_{0j}+\varepsilon_{ij}$$

Thirdly, we established a model including both the individual level and provincial level to test the effects of individual level variables and provincial level variables on cognitive scores of the older adults, concurrently. The provincial level variables included temperature ($\gamma_{01}$) and relative humidity ($\gamma_{02}$). In addition, a control variable was added, traffic condition ($\gamma_{03}$). Therefore, as shown in Equation (3), the cognitive score of each older adult individual ($Y_{ij}$) could be estimated by the average cognitive scores of the older adults in all provinces ($\gamma_{00}$) and the effects of individual level variables ($\gamma_{10},\;\gamma_{20}$, $\gamma_{30}$ and $\gamma_{40}$), provincial level variables ($\gamma_{01},\;\gamma_{02}$ and $\gamma_{03}$), and variances ($\upsilon_{0j}$ and $\varepsilon_{ij}$).
(3)$$\begin{array}{c} Y_{ij}=\gamma_{00}+\gamma_{10}age_{j}+\gamma_{20}gender_{j}+\gamma_{30}residence_{j}+\;\gamma_{40}chronic\;disease_{j}\\ +\gamma_{01}temperature_{j}+\gamma_{02}humidity_{j}+\gamma_{03}traffic_{j}+\upsilon_{0j}+\varepsilon_{ij} \end{array}$$

Finally, we established a full model, including main and moderating effects across levels. We tested the interaction between meteorological factors and demographic factors. As shown in Equation (4), ($\gamma_{11}$) represented the moderating effect of the provincial level variable (temperature) on the relationship between individual level variable (age) and the outcome variable (cognitive score). Similarly, ($\gamma_{12},\;\gamma_{13},\;\gamma_{21},\;\gamma_{22},\;\gamma_{23},\;\;\gamma_{31},\;\;\gamma_{32},\;\gamma_{33},\;\gamma_{41}$, $\gamma_{42}$ and $\gamma_{43}$) represented the moderating effects of provincial level variables on the relationship between individual level variables and the outcome variable.
(4)$$\begin{array}{l} Y_{ij}=\gamma_{00}+\gamma_{10}age_{j}+\gamma_{20}gender_{j}+\gamma_{30}residence_{j}\;+\;\gamma_{40}chronic\;disease_{j}\\ \;+\gamma_{01}temperature_{j}+\gamma_{02}humidity_{j}+\gamma_{03}traffic_{j}\\ \;+\gamma_{11}age_{j}\times temperature_{j}+\gamma_{12}age_{j}\times humidity_{j}+\gamma_{13}age_{j}\times traffic_{j}\\ \;+\gamma_{21}gender_{j}\times temperature_{j}+\gamma_{22}gender_{j}\times humidity_{j}+\gamma_{23}gender_{j}\times traffic_{j}\\ \;+\gamma_{31}residence_{j}\times temperature_{j}+\gamma_{32}residence_{j}\times humidity_{j}+\gamma_{33}residence_{j}\times traffic_{j}\\ \;+\gamma_{41}chronic\;disease_{j}\times temperature_{j}+\gamma_{42}chronic\;disease_{j}\times humidity_{j}\\ \;+\gamma_{43}chronic\;disease_{j}\times traffic_{j}+\upsilon_{0j}+\varepsilon_{ij} \end{array}$$

We took age, temperature, humidity, and highway construction as continuous variables. We set gender as a categorical variable, with male marked as 1 and female marked as 2. Chronic diseases (including diabetes, hypertension, and chronic cardiovascular disease) are divided into suffering (marked as 1) or not (marked as 0), and residence was distinguished by living in urban (marked as 1) or rural areas (marked as 2). IBM (Amonk, New York, NY, USA) SPSS statistics (v19.0) was used for data analysis.

## 3. Results

### 3.1. The Memory Test

#### 3.1.1. The Empty Model

The results in Table 2 show that there were significant differences in memory scores of the older adults from different provinces. We calculated intraclass correlation coefficient (ICC) considering both within-province variance and between-province variance. The ICC considering both within-province variance and between-province variance was equal to 0.072, which indicated that 7.2% of the memory test scores variation of the older adults was associated with provincial characteristics. The design effect was approximately equal to 1 + (average cluster size − 1) × intraclass correlation (ICC).

#### 3.1.2. The Individual Level Model

The individual level model assessed the association of the memory test scores with age, gender, residence, and chronic disease.

The results showed that older age (*γ*_10_ = −0.112, *t* = −13.464, *p* < 0.001), being male (*γ*_20_ = 0.311, *t* = 2.999, *p* = 0.003) and living in rural areas (*γ*_30_ = −0.582, *t* = −5.450, *p* < 0.001) were associated with lower memory test scores (Table 3 and Table 4).

The empty versus individual level models were assessed on the basis of the Restricted Maximum Likelihood (REML) test. Results showed that the individual model significantly improved the fit to the data compared to the empty model (Table 2).

#### 3.1.3. The Individual Level and Provincial Level Model

In this model, we investigated the effects of individual and provincial level variables on the memory test scores of older adults. As the results shown in Table 4, older age (*γ*_10_ = −0.112, *t* = −13.488, *p* < 0.001), being male (*γ*_20_ = 0.317, *t* = 3.053, *p* = 0.002) and living in rural areas (*γ*_30_ = −0.598, *t* = −5.602, *p* < 0.001) were associated with lower memory test scores, after adjustment for the provincial level variable.

Moreover, among the variables at the provincial level, humidity (*γ*_02_ = −0.048, *t* = −2.685, *p* = 0.008) was associated with memory scores of the older adults. Higher humidity was associated with lower memory test scores. The results of the REML test for the individual level and provincial level model are shown in Table 3.

#### 3.1.4. Interactive Effects Between Individual and Residential Variables on Cognitive Performance

As shown in Table 5, a positive moderating effect was found between the residence of the older adults and the average relative humidity of the provinces (*γ*_32_ = 0.045, *t* = 2.197, *p* = 0.028).

According to the humidity data in this study, we further divided the humidity into three levels, low-level humidity with less than 70% (480 cases), middle-level humidity with 70–79.9% (2110 cases), and high-level humidity with more than 80% (858 cases). Simple effect analysis showed that whether living in urban or rural areas, the memory scores of the older adults in high humidity were significantly lower (*p* < 0.001). The memory scores of the older adults living in urban areas decreased more significantly in high humidity than the older adults living in rural areas (*p* < 0.001) (shown in Figure 2). This result was still significant (*t* = 2.086, *p* = 0.37) when the other interactions were deleted from the model.

These results partially confirmed the H3; the relative humidity had a moderating effect on the relationship between residence (urban or rural) and cognitive performance of the older adults. The results of the REML test for the full model are shown in Table 2.

### 3.2. The Logical Sequence Test

#### 3.2.1. The Empty Model

The results in Table 6 show that there were significant differences in logical sequence scores of the older adults from different provinces. We calculated ICC considering both within-province variance and between-province variance. The ICC considering both within-province variance and between-province variance was equal to 0.061, which indicated that 6.1% of the logical sequence test scores variation of the older adults was associated with provincial characteristics.

#### 3.2.2. The Individual Level Model

The individual level model assessed the association of the logical sequence test scores with age, gender, residence, and chronic disease.

The results showed that older age (*γ*_10_ = −0.098, *t* = −9.269, *p* < 0.001), being female (*γ*_20_ = −1.037, *t* = −7.805, *p* < 0.001), and living in rural areas (*γ*_30_ = −0.736, *t* = −5.391, *p* < 0.001) were associated with lower logical sequence test scores (Table 7 and Table 8). However, the effect of suffering a chronic disease appeared to be positive (*γ*_40_ = 0.270, *t* = 1.981, *p* = 0.048).

The empty versus individual level models were assessed on the basis of REML test. Results showed that the individual model significantly improved the fit to the data compared to the empty model (Table 6).

According to Cohen (1988), when the ICC is less than 0.059, it is not necessary to use the mixed model [31,32]. In addition, as shown in Table 6, in the level 1 and 2 model and the full model, the ICC values were 0.055 and 0.049, respectively, indicating that the differences attributable to different provinces were too small. In addition to the size of ICC, in a logical sequence test, the design effect values of the level 1 and level 2 model and the full model were 8.53 and 7.71, respectively. The fitting degree of these models was too low to use the mixed model for further analysis. Therefore, we used a simpler single-layer linear regression analysis method., which was more convenient and the results were reliable. For the full model results, see Appendix A Appendix A.

## 4. Discussion

In this study including 3448 older adults, we used mixed effects models and found that living in rural areas and high-level humidity areas were associated with lower memory test performance (Table 7). Furthermore, we found that the strength of the negative association between the areas of residence and memory performance was moderated by humidity. In particular, this negative association was significantly more pronounced in urban areas compared to rural areas of residence. This study adds to the previous literature by revealing the association of individual and meteorological factors with cognitive function among older adults (Table 8).

The moderating effect of humidity on the relationship between residence and memory function in older age was illustrated by the association between high level humidity and memory function of urban older adults. Whether living in urban or rural areas, the memory scores of the older adults in high-humidity areas were significantly lower. This result is consistent with the studies by Vasmatzidis et al. and Trezza et al., which found that the memory function of people acutely exposed to high humidity is significantly lower than that of people exposed to low humidity [33,34]. In addition to memory function, some studies have found that a high relative humidity (70%) environment (e.g., subtropical climate) can reduce the cognitive ability of individuals [35], such as psychomotor speed and visual learning [33,35], making it more difficult for them to think clearly and reducing their alertness [36]. This may be due to the influence of relative humidity on the body’s ability of thermal sensation and thermoregulation. High humidity reduces the evaporation of human sweat, resulting in greater heat stress, which leads to the decline of cognitive function [33,37]. Older adults are less adaptable to climate change [38]. One possible explanation is that with adult aging, the regulatory capacity of older adults, such as cardiovascular function, gradually decreases, so the older adults are more vulnerable to the interference of heat stress [26,39]. Heat stress increases neuronal requirements for cognitive tasks in older adults [40,41] and disrupts brain functional connectivity [42,43]. Interestingly, we also found that the main effects of gender differences both in memory and logical sequence tests, which were consistent with previous views [44]. Although no significant interaction between gender and meteorological factors was found in this study, some studies suggested that due to the different physiological structures of males and females (such as weight and surface area), the adaptability to meteorological factors was also different [27,45]. It reminds us that in future cognitive interventions for the elderly, different functions need to focus on different gender groups.

Importantly, in this study, we found that in high humidity, the memory scores of the older adults living in urban areas were significantly lower than those living in rural areas. Previous studies have provided a possible explanation that long-term exposure to air pollution (e.g., particulate matter) is associated with cognitive impairment in the elderly [46,47,48]. Compared with rural areas, air pollution and other environmental problems in cities are more serious and meteorological factors such as high humidity play an important role in the elevation of air pollutant concentration [49]. Previous studies have shown that the impact of traffic conditions on the microenvironment is related to the distance from the residence to the highway [46]. This largely due to more intensive highway construction in cities; no matter in the design, construction, and operation stages, it will have a negative impact on the local micro ecological environment [9]. Moreover, high humidity conditions are more likely to cause pollution of atmospheric particles [50]. In addition, because of the urban heat island effect, a large number of studies have shown that in a high-heat and high-humidity environment, human cognitive function will be significantly reduced [34,35,51].

However, in the logical sequence test, we found that the performance differences of the older adults in different provinces were not significantly related to meteorological factors, but more related to the individual factors (such as age, gender, etc.). The results of previous studies are not consistent. Some studies show that temperature is related to mathematical ability, while others have not [52,53,54]. Another study points out that this inconsistency may be due to the lack of attention to gender differences in previous studies on temperature and mathematical ability [25]. At present, the research on the relationship between meteorological factors and mathematical ability, especially for older adults, needs to be verified by follow-up studies.

In this study, we used a mixed model to test the correlation between meteorological factors and cognitive function of older adults, and also examined the micro and macro multi-level effects. The limitations of this study were that we mainly investigated the influence of temperature and humidity on the cognitive function of older adults, while atmospheric pressure, precipitation, and other meteorological variables, as well as the air quality and the degree of greening in the city would also have an impact on the local microclimate. The meteorological data used in this study was accurate to the month, and we conceptualized the effects of humidity on cognitive outcomes as a contributor to a more chronic process of cognitive decline. In future research, using longitudinal datasets will help us to compare the acute exposure and cumulative effects of meteorological factors on cognitive function. In addition, we could further investigate the comprehensive effects of climate, environment, and ecology, using socio-economic status, formal education, home, and work environments (i.e., whether outfitted with air conditioners and dehumidifiers) as covariates or predictors in the equations to predict cognitive differences and report more accurate results.

## 5. Conclusions

In this study, 3448 older adults in CFPS were selected to assess the association of individual and meteorological factors with cognitive function by using mixed effects models. The main findings show that the negative association between humidity and memory performance is more pronounced in urban areas than in rural areas. This study explains how meteorological factors play a role in the influence of demographic factors on the cognitive function of older adults, and provides a reference for older adults to choose a more suitable residence of retirement.

## Figures and Tables

**Figure 1 ijerph-18-05981-f001:**
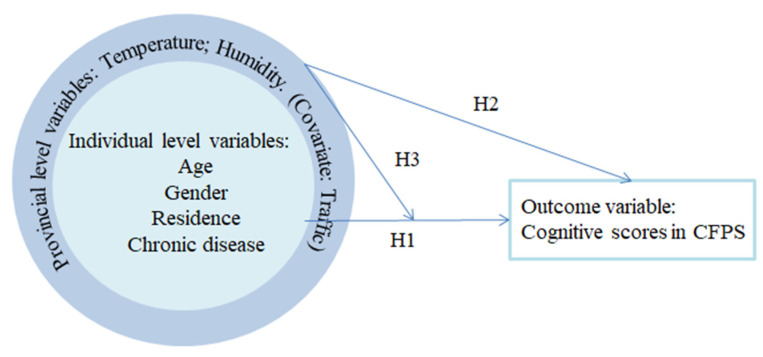
The proposed multi-level model.

**Figure 2 ijerph-18-05981-f002:**
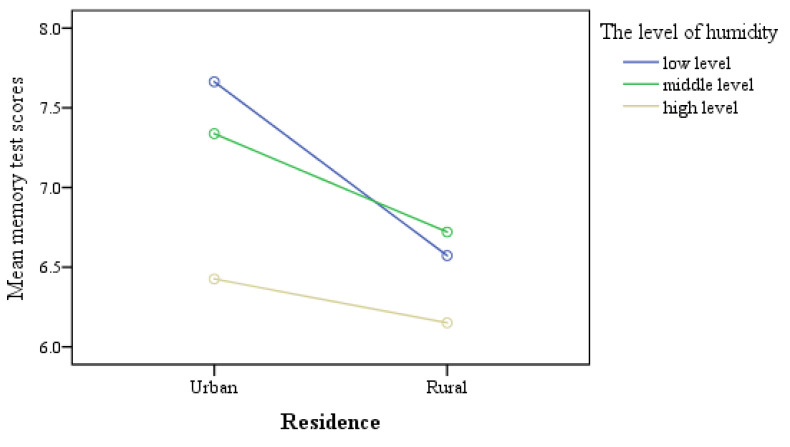
The memory scores between urban and rural areas under different humidity levels.

**Table 1 ijerph-18-05981-t001:** The demographic information of all the participants.

Variables	All Population	
*Continuous Variables*	n	Mean ± SD (min–max)
Age	3448	67.5 ± 6.0 (60–94)
Memory test score	3448	6.89 ± 3.02 (0–14)
Sequence test score	3448	4.59 ± 3.82 (0–14)
*Categorical variables*	n	%
Gender		
Male	2187	63.4
Female	1261	36.6
Education level		
Low (≤6)	1584	45.9
High (>6)	1864	54.1
Place of residence		
Urban	1903	55.1
Rural	1545	44.9
Chronic disease		
Yes	1081	31.3
No	2367	68.7

**Table 2 ijerph-18-05981-t002:** The gap value, variance, ICC, and design effect value of models.

Model	Gap (−2)	Variance		ICC	Design Effect
		Within-Province	Between-Province		
Empty model	17,314.501	8.740	0.679	0.072	10.86
Level 1 model	17,108.243	8.210	0.567	0.065	9.90
Level 1 and 2 model	17,134.779	8.203	0.503	0.058	8.94
Full model	17,180.733	8.200	0.514	0.059	9.08

**Table 3 ijerph-18-05981-t003:** Descriptive statistics of memory scores of categorical variables.

Categorical Variables	*n*	Mean ± SD
Gender		
male	2187	6.70 ± 2.955
female	1261	7.21 ± 3.104
Residence		
urban	1903	7.16 ± 3.089
rural	1545	6.55 ± 2.899
Chronic disease		
Yes	1081	6.79 ± 3.064
No	2367	6.93 ± 3.000

**Table 4 ijerph-18-05981-t004:** The main effects of the empty, individual level, and provincial level model.

Parameter	The Empty Model				The Individual Level Model				The Individual Level and Provincial Level Model			
	Estimate	*t*	*p*	*95% CI*	Estimate	*t*	*p*	*95% CI*	Estimate	*T*	*p*	*95% CI*
Intercept	6.738	37.758	<0.001	6.366–7.110	14.722	23.199	<0.001	13.477–15.966	17.889	11.935	<0.001	14.940–20.839
*Individual level variable*												
Age					−0.112	−13.464	<0.001	−0.128–−0.095	−0.112	−13.488	<0.001	−0.128–−0.096
Gender					0.311	2.999	0.003	0.108–0.515	0.317	3.053	0.002	0.113–0.520
Residence					−0.582	−5.450	<0.001	−0.791–−0.372	−0.598	−5.602	<0.001	−0.808–−0.389
Chronic disease					−0.078	−0.732	0.464	−0.287–0.131	−0.083	−0.781	0.435	−0.292–0.126
*Provincial level variable*												
Temperature									0.019	1.123	0.262	−0.014–0.053
Humidity									−0.048	−2.685	0.008	−0.084–−0.013
*Control variable*												
Traffic									9.49210 × 10 ^−7^	0.145	0.886	−1.28 × 10 ^−5^–1.47 × 10 ^−5^

**Table 5 ijerph-18-05981-t005:** The effects of individual level and provincial level variables in the full model.

Parameter	The Full Model				
	Estimate	*df*	*t*	*p*	*95% CI*
Intercept	22.273	3431.989	2.522	0.012	4.957–39.588
*Individual level variable*					
Age	−0.176	3419.325	−1.481	0.139	−0.409–0.057
Gender	2.482	3417.022	1.705	0.088	−0.373–5.337
Residence	−2.995	3312.824	−2.083	0.037	−5.815–−0.175
Chronic disease	1.105	3418.057	0.776	0.438	−1.687–3.898
*Provincial level variable*					
Temperature	0.007	3417.297	0.040	0.968	−0.326–0.340
Humidity	−0.101	3431.016	−0.828	−0.408	−0.341–0.139
*Control variable*					
Traffic	7.1501 × 10 ^−7^	19.059	0.108	0.915	−1.31 × 10 ^−5^–1.46 × 10 ^−5^
*Provincial level moderating effects*					
Age × Temperature	0.001	3422.119	0.437	0.662	−0.003–0.005
Age × Humidity	0.001	3419.433	0.310	0.757	−0.003–0.004
Gender × Temperature	−0.003	3423.568	−0.100	0.921	−0.058–0.052
Gender × Humidity	−0.028	3420.440	−1.381	0.167	−0.067–0.012
Residence × Temperature	−0.039	3309.418	−1.301	0.193	−0.099–0.020
Residence × Humidity	0.045	3258.090	2.197	0.028	0.005–0.085
Chronic disease × Temperature	−0.002	3421.325	−0.057	0.955	−0.060–0.056
Chronic disease × Humidity	−0.015	3416.835	−0.766	0.444	−0.054–0.024

**Table 6 ijerph-18-05981-t006:** The gap value, variance, ICC, and design effect value of models.

Model	Gap (−2)	Variance		ICC	Design Effect
		Within-Province	Between-Province		
Empty model	18,947.059	14.049	0.916	0.061	9.35
Level 1 model	18,806.274	13.452	0.835	0.058	8.94
Level 1 and 2 model	18,837.135	13.460	0.777	0.055	8.53
Full model	18,877.735	13.456	0.698	0.049	7.71

**Table 7 ijerph-18-05981-t007:** Descriptive statistics of logical sequence test scores of categorical variables.

Categorical Variables	*n*	Mean ± SD
Gender		
Male	2187	4.86 ± 3.87
Female	1261	4.11 ± 3.70
Residence		
Urban	1903	4.96 ± 3.97
Rural	1545	4.12 ± 3.59
Chronic disease		
Yes	1081	4.73 ± 3.88
No	2367	4.52 ± 3.80

**Table 8 ijerph-18-05981-t008:** The main effects of individual level variables.

Parameter	Estimate	*df*	*t*	*p*	*95% CI*	
*Individual* *level variable*						
Intercept	13.632	2218.282	16.831	<0.001	12.044	15.221
Age	−0.098	3431.986	−9.269	<0.001	−0.119	−0.078
Gender	−1.037	3431.832	−7.805	<0.001	−1.297	−0.776
Residence	−0.736	3404.812	−5.391	<0.001	−1.003	−0.468
Chronic disease	0.270	3431.226	1.981	0.048	0.003	0.538

## Data Availability

The data presented in this study are openly available in the following websites: (1) The China Family Panel Studies (http://isss.pku.edu.cn); (2) The National Meteorological Information Center of China (http://data.cma.cn); (3) The National Bureau of statistics of China (http://www.stats.gov.cn).

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
