# Peer review of "The Association of Meteorological Factors with Cognitive Function in Older Adults"

_ijerph, 2021, doi:10.3390/ijerph18115981_

Round 1
Reviewer 1 Report
Thank you for the opportunity to review this manuscript, which describes the association between meteorological factors and cognitive functions among older adults in China, using mixed models to capture both individual level and province level factors. I found this paper to be interesting and generally well written. Please see below my general and specific comments.
General Comments
1. The criteria for deciding that a mixed model (including provincial level effects) should be used for the memory test but not for the logical processing test appears to be based solely on the ICC. Please provide further justification and a threshold level of between province-explained variance that would be required in order to determine that a mixed model was feasible.
2. For the logical processing model why does the individual level effects model only deal with main effects? Were second-order interaction terms tested? I would be interested in knowing whether chronic disease was equally distributed across gender, and whether there may be an interaction between gender and chronic disease in logical processing.
3. Given the explanation proposed for the interaction between humidity and urban/rural residence (elevating air pollution concentration) it would be helpful for the authors to give some discussion of the control variable (traffic) as assessed by highway density. At present this is used in the model but its suitability as a control variable in the context of the interpretation of the results is not discussed.
4. The limitations section could be more detailed. At present it is not clear whether the authors conceptualise the effects of humidity on cognitive outcomes as a transitory effect (ie acute and short term) or a contributor to a more chronic process of cognitive decline. The limitations section could raise this issue and discuss methodological approaches for future studies.
Specific Comments
Table 4: some p values are shown with a zero preceding the decimal point, others are not. Please adopt a consistent format
Line 296: The effect of suffering a chronic disease appears to be positive, although the finding is presented as though it is associated with lower logical processing scores. Please clarify if this is correct, and/or if the presentation of the levels of the chronic disease variable (1 = disease, 0 = no disease) are correct. Also the descriptive results (means) in Table 7 suggest that presence of chronic disease is associated with higher scores.
Line 320: In text the authors refer to Vombatids et al however the citation at the end of the sentence refers to Vazmatzidis et al. There is no Vombatids et al in the reference list.
Line 340: Colloquial language "as we all know" and "damage the emotions..." may not accurately reflect the findings that are being referenced here
Reviewer 2 Report
In this study the authors use mixed effects models to assess the correlation between meteorological factors and cognitive function in older adults from different regions in China. Overall, the authors present an interesting correlation that should be studied further. The study presents a comprehensive and well-designed methodology, with data from more than 3400 individuals from different regions. The authors correctly identify the limitations of the study. Therefore, I consider the article of great interest to be published in the IJERPH journal. Below I include a number of specific comments that authors should keep in mind.
In addition, the authors should correct some small errors:
The introduction is complete and addresses the topic clearly. Statements in the text must be accompanied by references (line 62). why have the authors chosen highway construction as a covariate? The authors should explain the selection of this particular variable.
Methodology. The authors use an accurate methodology for the study, where they evaluate different variables. With respect to these variables, some questions arise: What does educational level < or > 6 mean? In Table 1, cardiovascular diseases appear as a variable, whereas in the rest of the work chronic diseases appear, is this correct?
Results: Error in Figure 2: in the graph, change Ubran for Urban.
Review the references used because sometimes the format is not the same in the text. For example: Tian et al., 2020 (pos 326) & Tian, Fang, & Liu, 2020 (pos 325).
Finally, the discussion, although well presented, is very brief and the authors should revise and apply it. It is interesting the difference observed in the results of the memory tests as a function of gender. The authors should discuss this fact in greater depth.
Once these small details have been corrected, I recommend the publication of this interesting article in the IJERPH journal.
